# Machine learning health-related applications in low-income and middle-income countries: a scoping review protocol

Rodrigo M Carrillo-Larco ![ORCID],[1,2] Lorainne Tudor Car,[3,4] Jonathan Pearson-Stuttard,[5] Trishan Panch,[6] J Jaime Miranda ![ORCID],[2,7] Rifat Atun[8]

## ABSTRACT

**Introduction** Machine learning (ML) has been used in bio-medical research, and recently in clinical and public health research. However, much of the available evidence comes from high-income countries, where different health profiles challenge the application of this research to low/middle-income countries (LMICs). It is largely unknown what ML applications are available for LMICs that can support and advance clinical medicine and public health. We aim to address this gap by conducting a scoping review of health-related ML applications in LMICs.

**Methods and analysis** This scoping review will follow the methodology proposed by Levac *et al*. The search strategy is informed by recent systematic reviews of ML health-related applications. We will search Embase, Medline and Global Health (through Ovid), Cochrane and Google Scholar; we will present the date of our searches in the final review. Titles and abstracts will be screened by two reviewers independently; selected reports will be studied by two reviewers independently. Reports will be included if they are primary research where data have been analysed, ML techniques have been used on data from LMICs and they aimed to improve health-related outcomes. We will synthesise the information following evidence mapping recommendations.

**Ethics and dissemination** The review will provide a comprehensive list of health-related ML applications in LMICs. The results will be disseminated through scientific publications. We also plan to launch a website where ML models can be hosted so that researchers, policymakers and the general public can readily access them.

For numbered affiliations see end of article.

**Correspondence to**
Rodrigo M Carrillo-Larco;
r.carrillo-larco@imperial.ac.uk

## INTRODUCTION

Machine learning (ML) refers to the process through which computers, models or algorithms, learn and improve from data and processes, rather than from specific programmed instructions.[1 2] ML can be used in tasks such as classification (eg, whether a tumour is benign or malign based on patterns or characteristics), clustering (eg, group patients with similar profiles for targeted prevention or treatment interventions) or prediction (eg, forecast propensity to risk or

probability of outcome of a disease following interventions).[2] Of note, there could be overlap across different tasks.

### Strengths and limitations of this study

► The search strategy is informed by a solid framework.
► The search will be conducted in several information sources.
► The screening/review process will be performed by two reviewers independently.
► We will not search abstracts or conference proceedings specialised in machine learning.
► We will focus on information sources, where health-related outcomes are most likely to be found.

ML is widely used in bio-medical sciences, and more recently in clinical and public health research.[3 4] Systematic reviews on health-related applications of ML have explored questions such as the accuracy of ML for diagnosis or outcome prediction,[5–7] but most of the research studies included in these reviews have come from high-income countries and the findings may not apply to low/middle-income countries (LMICs) because of the variability in access to healthcare and difference in the disease burden. It is largely unknown what ML applications are available for LMICs that can support and advance clinical medicine and public health.

Our aim was to address this gap in the evidence by conducting a scoping review of health-related applications of ML in LMICs to synthesise published evidence and to garner lessons to inform further research and policy development. The review will provide the first comprehensive list of health-related ML applications in LMICs.

## METHODS
### Overview
This is a scoping review of the published scientific literature. This protocol adheres to the Preferred Reporting Items for Systematic Reviews and Meta-Analyses Protocols (PRISMA-P) guidelines (online supplementary material S1),[8] and the methodology follows the procedures suggested by Levac *et al*.[9] The final publication of this work will adhere to the PRISMA extension Scoping Reviews (PRISMA-ScR) recommendations.[10]

PRISMA-P is a standard method to report review protocols; similarly, PRSIMA-ScR is a well-known instrument to guide the reporting of scoping reviews. The methodology proposed by Levac and colleagues is useful for scoping reviews that aim to generate a broad picture of the available evidence on a subject.

Understanding that a key feature of scoping reviews is a broad research question, we aim to answer: what have been the health-related applications of ML techniques in LMICs? This review will summarise scientific evidence on research that have used data from LMICs for ML applications in clinical medicine (eg, risk prediction for clinical decisions) and public health (eg, vector control in an endemic area) to provide solutions to health problems in LMICs.

### Definitions
For this work, we will follow the following definitions:
▶ LMICs: classification by the World Bank country income grouping (see search strategy in table 1 for a list of countries).[11]
▶ ML: analytical techniques through which computers learn directly from data, examples and experiences, rather than from a pre-programmed rule.[1 2]
▶ ML techniques: analytical methods within the ML remit, that is, where machines have learnt from data or processes through ML techniques or algorithms. These include, but are not limited to[2 12]: support vector machine, support vector regression, decision trees, random forest, neural network, Bayesian network, artificial neural network, computer vision systems, computer-assisted image processing and natural language processing.

### Eligibility criteria
We will include scientific evidence that meets the following inclusion criteria (figure 1).

### Primary research studies
We will include experimental and observational studies that have used and reported quantitative data. We will also screen relevant systematic reviews or narrative reviews for eligible primary studies and include them in our review if relevant. We will not include qualitative studies, opinions, conference abstracts, letters, editorials or any other scientific work in which data have not been actively analysed within an ML framework.

### Machine learning research should have used data from LMICs, that is, machine learning research that used solely data from LMICs
We will focus on ML research based on data from LMICs applied to LMICs. We will not include ML research that used data from high-income countries even though it could have been applied to a LMIC, and neither studies using LMICs data applied to high-income countries. This scoping review focuses on the applications of ML techniques in LMICs; these applications should have been developed using LMICs data, because prediction models and other algorithms work better, that is, have better accuracy, in the populations—data—for which they were developed. Conversely, when these models are applied to other populations, settings or data, they need some modification (eg, recalibration). We will also exclude studies that have used LMIC data in a consortium or data pooling group if the model or results cannot be separated for the LMIC alone; in other words, we will exclude a report if this used LMIC data in aggregate with data from high-income countries, but the application cannot be separated for the LMIC alone.

Models developed in sites other than LMICs may not work correctly in these countries. For example, projects with digital imaging from high-income countries may reflect a different scenario; that is, images from streets in LMICs may depict objects or features not found in high-income countries. Another example could be projects for sound/noise classification. Those from high-income countries may not identify the variety of noises usually available in LIMCs (eg, loud cars or indiscriminate use of car horns). Finally, LMICs still have sizeable rural areas with large populations. Extrapolating models built for highly urbanised cities may not be adequate for rural sites.

Examples of studies of interest include: (1) development of a 'deep learning-based visual evaluation algorithm' to early identify cervical cancer signs based on data from women in Costa Rica,[13] (2) classification of free-text (random forest) in emergency department records from nine hospitals in Nicaragua[14] and (3) automatic classification (neural networks) of paediatric pneumonia based on ultrasound records from children in Peru.[15]

Distinguishing between ML applications and more conventional statistical methods could be challenging because in some cases the definitions are unclear, for instance, regression analysis. Nonetheless, from the context of the scientific paper, from the aims or overall methodological approach, it is possible to reckon whether a study uses ML techniques versus more conventional statistical methods. If needed, we would reach out to authors for further information.

### The outcome of the study/analysis was to improve a health-related outcome
The *primary outcome* of the selected studies should have sought improvements in the following health-related endpoints along the care cascade: diagnosis, treatment, control, survival, complications and mortality. These outcomes have been selected because of their relevance in healthcare, clinical medicine and public health. As

| Table 1 | Overall search terms |
|---------|---------------------|
| 1 | artificial intelligence.mp. |
| 2 | exp Artificial Intelligence/ |
| 3 | machine learning.mp. |
| 4 | exp machine learning/ |
| 5 | deep learning.mp. |
| 6 | unsupervised machine learning.mp. |
| 7 | supervised machine learning.mp. |
| 8 | computational Intelligence.mp. |
| 9 | predictive analytic*.mp. |
| 10 | support vector machine.mp. |
| 11 | support vector regression.mp. |
| 12 | decision tree*.mp. |
| 13 | random forest.mp. |
| 14 | neural network*.mp. |
| 15 | exp Neural Networks/ |
| 16 | bayesian network*.mp. |
| 17 | artificial neural network*.mp. |
| 18 | convolutional neural network*.mp. |
| 19 | computer vision systems.mp. |
| 20 | exp Image Processing, Computer-Assisted/ |
| 21 | natural language processesing.mp. |
| 22 | 1 or 2 or 3 …or 21 |
| 23 | (("Afghanistan") or ("Benin") or ("Burkina Faso") or ("Burundi") or ("Central African Republic") or ("Chad") or ("Comoros") or ("Democratic Republic of the Congo") or ("Eritrea") or ("Ethiopia") or ("Gambia") or ("Guinea") or ("Guinea-Bissau") or ("Haiti") or ("Democratic People's Republic of Korea") or ("Liberia") or ("Madagascar") or ("Malawi") or ("Mali") or ("Mozambique") or ("Nepal") or ("Niger") or ("Rwanda") or ("Senegal") or ("Sierra Leone") or ("Somalia") or ("South Sudan") or ("Tanzania") or ("Togo") or ("Uganda") or ("Zimbabwe") or ("Armenia") or ("Bangladesh") or ("Bhutan") or ("Bolivia") or ("Cape Verde") or ("Cambodia") or ("Cameroon") or ("Congo") or ("Cote d'Ivoire") or ("Djibouti") or ("Egypt") or ("El Salvador") or ("Ghana") or ("Guatemala") or ("Honduras") or ("India") or ("Indonesia") or ("Kenya") or ("Micronesia") or ("Kosovo") or ("Kyrgyzstan") or ("Laos") or ("Lesotho") or ("Mauritania") or ("Moldova") or ("Mongolia") or ("Morocco") or ("Myanmar") or ("Nicaragua") or ("Nigeria") or ("Pakistan") or ("Papua New Guinea") or ("Philippines") or ("Samoa") or ("Atlantic Islands") or ("Melanesia") or ("Sri Lanka") or ("Sudan") or ("Swaziland") or ("Syria") or ("Tajikistan") or ("Timor-Leste") or ("Tonga") or ("Tunisia") or ("Ukraine") or ("Uzbekistan") or ("Vanuatu") or ("Vietnam") or ("Middle East") or ("Yemen") or ("Zambia") or ("Albania") or ("Algeria") or ("American Samoa") or ("Angola") or ("Argentina") or ("Azerbaijan") or ("Republic of Belarus") or ("Belize") or ("Bosnia and Herzegovina") or ("Botswana") or ("Brazil") or ("Bulgaria") or ("China") or ("Colombia") or ("Costa Rica") or ("Cuba") or ("Dominica") or ("Dominican Republic") or ("Equatorial Guinea") or ("Ecuador") or ("Fiji") or ("Gabon") or ("Georgia") or ("Grenada") or ("Guyana") or ("Iran") or ("Iraq") or ("Jamaica") or ("Jordan") or ("Kazakhstan") or ("Lebanon") or ("Libya") or ("Macedonia (Republic)") or ("Malaysia") or ("Indian Ocean Islands") or ("Mauritius") or ("Mexico") or ("Montenegro") or ("Namibia") or ("Palau") or ("Panama") or ("Paraguay") or ("Peru") or ("Romania") or ("Russia") or ("Serbia") or ("South Africa") or ("Saint Lucia") or ("Saint Vincent and the Grenadines") or ("Suriname") or ("Thailand") or ("Turkey") or ("Turkmenistan") or ("Venezuela") or (developing countr) or (low-income countr*) or (middle-income countr*) or (low-middle income countr*) or (upper-middle income countr*)) |
| 24 | 22 and 23 |
| 25 | exp animals/ not humans.sh. |
| 26 | 24 not 25 |
| 27 | Remove duplicates from 26 |

*secondary outcome*, we will also include studies that have reported endpoints related to cost, efficiency and productivity in the healthcare process.

## Information sources

We will conduct the search in three databases through Ovid: Embase, Medline and Global Health; in addition, we will search Cochrane and Google Scholar (first 10 pages). Besides Google Scholar, no other grey literature source will be searched. These sources will be used without language or time/year restrictions. We have a very diverse team covering many languages and we will tap into our networks at our institutions in case we come across a study in a language that the authors do not speak.

## Search strategy

Based on recent systematic reviews of ML applications on health-related endpoints: outcome prediction in gastrointestinal bleeding,[7] assessment of physicians knowledge,[16] applications on genomic data to predict outcomes in

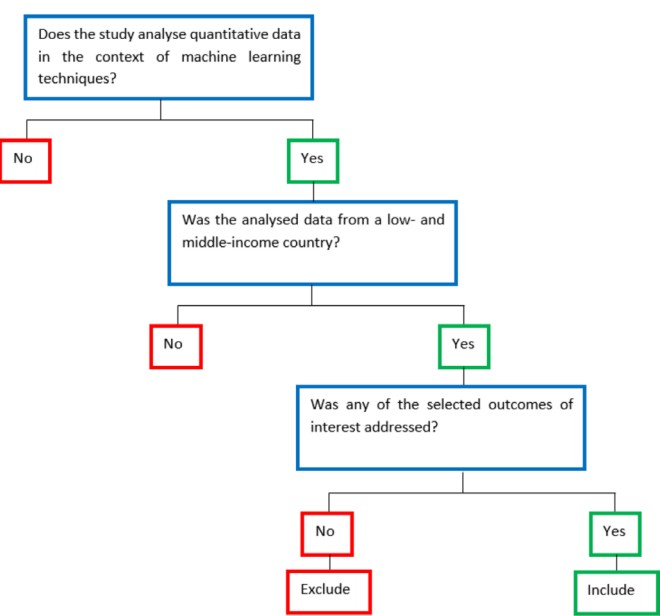

**Figure 1** Algorithm to filter search results.

cancer patients[17] and available ML algorithms to improve genomic data analysis,[18] we developed our search strategy (table 1). We will screen the references of the included reports for any other relevant studies.

## Study records

Results will be downloaded into EndNote, where duplicates will be omitted, and a second cleaning of duplicated results will be conducted using the online tool Rayyan.[19] Titles and abstracts will be reviewed by two independent researchers following the above detailed selection criteria (figure 1); discrepancies will be solved by consensus or by a third party. Selected titles will be sought in full text, which will be assessed by two independent reviewers following the same selection criteria (figure 1). Again, in case of conflicts, these will be solved by consensus or by a third party.

## Data collection

The reviewers will decide on a list of items that will be extracted to successfully answer the research question. These items will be implemented in an Excel spreadsheet before data extraction and will not be modified afterwards. Information of interest includes: country of origin; analytical approach; type of data used; data source; model performance; outcome of interest; number of observations and whether the model is available for independent use or reproduction. Data from the selected reports will be extracted by two researchers independently; if there were discrepancies, these will be solved by consensus among them or by a third party.

## Risk of bias in individual studies

Because this is a scoping review aiming to summarise available evidence to identify research gaps and potential uses of ML to improve health-related outcomes in LMICs, no risk of bias of individual studies is planned.

## Data synthesis

We anticipate a large heterogeneity of selected reports, both in terms of methodology and outcomes, as well as target population and data sources. Therefore, no meta-analysis is planned and only a qualitative summary will be conducted. Following current recommendations for scoping reviews and evidence mapping,[20] we will present the results through tables and figures, for example, a map pointing out where studies have been conducted and summarising key characteristics. As needed, we will consider other figures such as a matrix evidence.[20]

## Ethics and dissemination

This scoping review will not require ethical approval because it did not study human subjects; also, it included sources that are or can be made available to the public.

We plan to report our findings in a scientific publication. In addition, and depending on available resources, we aim to produce a website (or implement in an existing website) in which the findings and summarised reports can be easily accessed. Furthermore, we aim to host the ML models so that researchers, policymakers and the general public can readily access them. Where the ML models are open access or can be accessed through the original reports, these will be hosted on the website or a link to the original source will be provided; conversely, where ML models are not open access, we will contact the study authors and ask for the model to be hosted in our website or for a link to their ML model. This dissemination plan aims to increase visibility of ML research in LMICs and to increase the use of available models, thereby encouraging further research to improve health outcomes. We will engage with the communication office in our universities to promote this website through relevant channels, including but not limited to social media, newsletters and institutional websites.

## Patient and public involvement

No patients will be directly involved in the design, planning and conception of this study.

**Author affiliations**
¹Department of Epidemiology and Biostatistics, School of Public Health, Imperial College London, London, UK
²CRONICAS Centre of Excellence in Chronic Diseases, Universidad Peruana Cayetano Heredia, Lima, Peru
³Family Medicine and Primary Care, Lee Kong Chian School of Medicine, Nanyang Technological University, Singapore
⁴Department of Primary Care and Public Health, School of Public Health, Imperial College London, London, UK
⁵Department of Epidemiology and Biostatistics and MRC-PHE Centre for Environment and Health, School of Public Health, Imperial College London, London, UK
⁶Wellframe Inc, Boston, Massachusetts, USA
⁷Facultad de Medicina "Alberto Hurtado", Universidad Peruana Cayetano Heredia, Lima, Peru
⁸Harvard T.H Chan School of Public Health and Harvard Medical School, Harvard University, Cambridge, Massachusetts, USA

**Contributors** RMC-L conceived the idea and drafted the manuscript. JP-S, TP and RA provided advice to improve the research question and LTC to improve the

protocol. JJM and RA edited and provided insights to improve the protocol. All authors approved the submitted version.

**Funding** RMC-L has been supported by a Strategic Award, Wellcome Trust-Imperial College Centre for Global Health Research (100693/Z/12/Z) and Imperial College London Wellcome Trust Institutional Strategic Support Fund (Global Health Clinical Research Training Fellowship) (294834/Z/16/Z ISSF ICL). RMC-L is supported by a Wellcome Trust International Training Fellowship (214185/Z/18/Z). The funders had no role in this work and decision to submit for publication.

**Competing interests** None declared.

**Patient and public involvement** Patients and/or the public were not involved in the design, or conduct, or reporting, or dissemination plans of this research.

**Patient consent for publication** Not required.

**Provenance and peer review** Not commissioned; externally peer reviewed.

**Open access** This is an open access article distributed in accordance with the Creative Commons Attribution 4.0 Unported (CC BY 4.0) license, which permits others to copy, redistribute, remix, transform and build upon this work for any purpose, provided the original work is properly cited, a link to the licence is given, and indication of whether changes were made. See: https://creativecommons.org/licenses/by/4.0/.

**ORCID iDs**
Rodrigo M Carrillo-Larco http://orcid.org/0000-0002-2090-1856
J Jaime Miranda http://orcid.org/0000-0002-4738-5468

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
