## [Reviewer comments · BMJ Open]

ARTICLE DETAILS

TITLE (PROVISIONAL)	Machine learning health-related applications in low- and middle-income countries: A scoping review protocol
AUTHORS	Carrillo-Larco, Rodrigo; Tudor Car, Lorraine; Pearson-Stuttard, Jonathan; Panch, Trishan; Miranda, J. Jaime; Atun, Rifat

VERSION 1 – REVIEW

REVIEWER	Alexandre Chiavegatto Filho School of Public Health of the University of São Paulo, Epidemiology
REVIEW RETURNED	18-Dec-2019

GENERAL COMMENTS	The article is a study protocol of a future review of health-related machine learning applications in low-and-middle-income countries (LMICs). While the area of research is very important and growing substantially, it is hard to understand the point of publishing a study protocol when it is just a simple review and there are no hypotheses to be tested. The only justification I can think of would be if it was a complicated and detailed analysis, but this is a very straightforward literature review that can be easily included in a Methods section of the future manuscript. Minor considerations: - In the Introduction, the authors mention classification and prediction as separate machine learning tasks, but classification is a prediction task.- The authors should include publications from conferences specialized in machine learning, as it is how most studies in computer science are published.
--

REVIEWER	George Gotsadze Curatio International Foundation
REVIEW RETURNED	07-Jan-2020

GENERAL COMMENTS	Overall this would be an interesting study and thanks to authors for embarking on the protocol development and its publication for the benefit of others. I have some points for authors to consider before publishing the paper: 1. Authors state that most studies about ML were produced using data from High-income settings and therefore suggest to focus on LMIC setting in their scoping review, which is a welcome development. However, justification of why some ML applications (such as those using digital imaging for example) developed in
--

	high-income settings may not be relevant to LMICs needs further elaboration, which has to also support the restrictions presented on page 8 line 13. "We will not include machine learning research that used data from high-income countries even though it could have been applied to an LMIC, and neither studies using LMICs data applied to high-income countries." 2. Authors plan to use English search terms but they do not intend to restrict the language of identified papers (page 9 line 15). It is not clear (a) if English search terms would produce papers developed in other languages and (b) if authors will be able to review the papers if they are in Russian or Chinese and (c) how they will deal with the language in general. 4. The data collection section (page 9, Line 49) needs more elaboration. In its current presentation, it is a rather high level and a more nuanced description will be helpful for the reproducibility of the research protocol. 5. Developing a website/repository of ML applications relevant to LMIC settings could be a useful first step for effective dissemination, but not sufficient unless the website/repository itself if actively promoted using various communication channels. Thus, authors are recommended to re-think and further elaborate on how the website could be promoted.
--	---

REVIEWER	Andreas Triantafyllidis Centre for Research & Technology Hellas, Greece
REVIEW RETURNED	22-Jan-2020

GENERAL COMMENTS	The authors do not report the reasons behind the use of selected reporting tools, i.e., PRISMA-P, Levac, PRISMA-ScR. Proper justification is needed for the selection of those tools. Please revise what machine learning techniques are: Computer vision systems, computer-assisted image processing and natural language processing, are considered by the authors as machine learning techniques, but they actually are broader scientific fields. In eligibility criteria, authors don't mention the publication period of the studies screened. If there is no restriction on the date, a study should be published, this should be clearly reported. Authors don't also mention the dates that their search will take place. The sentence "This scoping review focuses on the applications of machine learning techniques in LMICs developed using LMICs data as prediction models and other algorithms work better, i.e. have better accuracy, in the population – data – for which they have been developed. Conversely, when these models are applied to other populations, settings or data, they need some modification (e.g., re-calibration).", is not understandable. The authors state: "The primary outcome of the selected studies should have sought improvements in the following health-related endpoints along the care cascade: diagnosis, treatment, control, survival, complications and mortality." However, prognosis, or prediction, is also an important area of ML applications, which is not clearly included as a study outcome.
--

	Justification for the use of tool Rayyan is needed. Search dates and reporting on whether the grey literature will be explored, are needed. Systematic reporting of risk of bias is not planned, however this should be elaborated even through discussion/limitations, in the scoping review, because handling of bias is an important dimension in conducted studies. Key characteristics of the included studies for data synthesis, are not presented. E.g., ML technique used, number of participants, ML outcome, implication for clinical practice, etc. Errors in written English: “have used and report”, “We will synthesize”, “In addition, an depending on available resources,” etc.
--	---

REVIEWER	Michael R Barnes Queen Mary University of London, UK
REVIEW RETURNED	04-Feb-2020

GENERAL COMMENTS	This is an interesting scoping review proposal which is quite clearly described. However as it currently stands I do not think it will be a comprehensive review, some adjustments are necessary. See my comments below. Comments 1) The definition of machine learning (ML) “analytical techniques through which computers learn directly from data, examples and experiences, rather than from a pre-programmed rule” is good. However distinguishing ML from more conventional statistical methods could be challenging and in some cases the definitions are rather unclear. For example regression analysis could be used as a machine learning approach but in some cases might be considered a pre-programmed rule. 2) p6, l10 - Arguably classification and clustering are the same processes. 3) Table 1. lists a number of terms, but is not comprehensive, authors should also consider regression analysis, gradient boosting, genetic algorithms, naïve Bayes, ridge regression, the list goes on... Some online resources provide comprehensive lists of methods. A useful output of the systematic review would be a comprehensive list of ML search terms. More generic terms like “training model” might also be used to detect ML studies. 4) Have the authors considered a specific review of clinical trials performed in LMICs? 5) “we will search Cochrane and Google Scholar (first 10 pages).” – why the first ten pages, this will surely not be a comprehensive approach. Searching google scholar with the term (“machine learning” and LMIC) there are 498 results covering more than 10 pages.
--

VERSION 1 – AUTHOR RESPONSE

Reviewer #1

Q1: While the area of research is very important and growing substantially, it is hard to understand the point of publishing a study protocol when it is just a simple review and there are no hypotheses to be tested.

A1: We appreciate the positive feedback. Publishing the protocol is not only relevant for complex studies or where hypotheses are being tested; it also has to do with transparency and informing the scientific community about the ongoing work. Given how novel these machine learning techniques may be in low- and middle-income countries, paired with the fact that evidence for these countries is spread out, we believe the protocol is still relevant and worth publishing. Furthermore, publishing protocols of scoping reviews is a standard practice, and there are plenty of examples published by BJM Open:

- <https://bmjopen.bmj.com/content/10/2/e034301>
- <https://bmjopen.bmj.com/content/10/2/e034032>
- <https://bmjopen.bmj.com/content/10/2/e032912>
- <https://bmjopen.bmj.com/content/10/2/e033592>

Q2: In the Introduction, the authors mention classification and prediction as separate machine learning tasks, but classification is a prediction task.

A2: We understand these may not be separate tasks. However, in the introduction these are separated to include examples in which these methods have been used. They are not separated to suggest that these are completely different approaches; conversely, this classification does not mean that there is no overlap across these tasks. The text in the introduction reads: “*Machine learning can be used in tasks such as classification (e.g., whether a tumour is benign or malign based on patterns or characteristics), clustering (e.g., group patients with similar profiles for targeted prevention or treatment interventions) or prediction (e.g., forecast propensity to risk or probability of outcome of a disease following interventions).*” These lines do not aim to argue that these tasks are different, but they are umbrella terms to group examples of previous works. To further address this issue in the manuscript, a new line has been introduced at the end of the first paragraph in the introduction: “*Of note, there could be overlap across different tasks*”.

Q3: The authors should include publications from conferences specialized in machine learning, as it is how most studies in computer science are published.

A3: We do in fact include conference proceedings through Google Scholar and Global Health database.

Reviewer #2

Q1: Authors state that most studies about ML were produced using data from High-income settings and therefore suggest to focus on LMIC setting in their scoping review, which is a welcome development. However, justification of why some ML applications (such as those using digital imaging for example) developed in high-income settings may not be relevant to LMICs needs further elaboration, which has

to also support the restrictions presented on page 8 line 13. "We will not include machine learning research that used data from high-income countries even though it could have been applied to an LMIC, and neither studies using LMICs data applied to high-income countries."

A1: We appreciate the positive feedback. We have further elaborated as requested. The new lines read: *"Models developed in sites other than LMICs may not work correctly in these countries. For example, projects with digital imaging from high-income countries may reflect a different scenario; i.e., images from streets in LMICs may depict objects or features not found in high-income countries. Another example could be projects for sound/noise classification. Those from high-income countries may not identify the variety of noises usually available in LMICs (e.g., loud cars or indiscriminate use of car horns). Finally, LMICs still have sizable rural areas with large populations. Extrapolating models built for highly urbanized cities may not be adequate for rural sites"*.

Q2: Authors plan to use English search terms but they do not intend to restrict the language of identified papers (page 9 line 15). It is not clear (a) if English search terms would produce papers developed in other languages and (b) if authors will be able to review the papers if they are in Russian or Chinese and (c) how they will deal with the language in general.

A2: It is correct that we will only use search terms in English; these terms will still produce papers in other languages. We will include papers in all languages that have a title and an abstract in English. We have a very diverse team covering many languages and we will tap into our networks at our institutions in case we come across a study in a language that the authors do not speak. This last remark has also been included in the 'information sources' section of the manuscript.

Q3: The data collection section (page 9, Line 49) needs more elaboration. In its current presentation, it is a rather high level and a more nuanced description will be helpful for the reproducibility of the research protocol.

A3: This section has been further elaborated with the following lines: *"Information of interest includes: country of origin; analytical approach; type of data used; data source; model performance; outcome of interest; number of observations; and whether the model is available for independent use or reproduction."*

Q4: Developing a website/repository of ML applications relevant to LMIC settings could be a useful first step for effective dissemination, but not sufficient unless the website/repository itself is actively promoted using various communication channels. Thus, authors are recommended to re-think and further elaborate on how the website could be promoted.

A4: This is indeed a useful comment and suggestion, which we will keep in mind. We have added the following lines: *"We will engage with the communication office in our universities to promote this website through relevant channels including but not limited to social media, newsletters and institutional websites."*

Reviewer #3

Q1: Errors in written English: “have used and report”, “We will synthesize”, “In addition, an depending on available resources,” etc.

A1: These typos have been corrected and the text proofread.

Q2: Key characteristics of the included studies for data synthesis, are not presented. E.g., ML technique used, number of participants, ML outcome, implication for clinical practice, etc.

A2: This information has been further elaborated: “*Information of interest includes: country of origin; analytical approach; type of data used; data source; model performance; outcome of interest; number of observations; and whether the model is available for independent use or reproduction*”.

Q3: Systematic reporting of risk of bias is not planned, however this should be elaborated even through discussion/limitations, in the scoping review, because handling of bias is an important dimension in conducted studies.

A3: Risk of bias assessment is not part of the standard methodology for *scoping reviews*, unlike systematic reviews. Given the broad scope of a scoping review, particularly of ours in which we try to generate an overall picture of what evidence there is on health and machine learning in low- and middle-income countries, we will rather not conduct a risk of bias assessment. Moreover, risk of bias is evaluated with specific tools/questionnaires, none of which -to the best of our knowledge- is specific for the kind of studies we might retrieve.

Q4: Justification for the use of tool Rayyan is needed. Search dates and reporting on whether the grey literature will be explored, are needed.

A4: Rayyan is an online tool to conduct reviews; there are other software for this purpose: RevMan or Covidence. Rayyan is a free and friendly software that allows collaborative work. The authors are familiar and have used this software in previous reviews. The search will be conducted according to time availability of the researchers. We will present the date of our searches in the final review. Besides Google Scholar, no other grey literature source will be searched; this last remark has been included in the ‘Information sources’ section of the protocol. We will acknowledge this as a potential limitation in the final report.

Q5: The authors state: “The primary outcome of the selected studies should have sought improvements in the following health-related endpoints along the care cascade: diagnosis, treatment, control, survival, complications and mortality.” However, prognosis, or prediction, is also an important area of ML applications, which is not clearly included as a study outcome.

A5: We fully agree. In the introduction of the manuscript, we state that both prognosis and prediction will be included in the review, and they are inherent to the listed outcomes. For example, survival outcomes will require prediction analysis. Similarly, treatment outcomes will require a prognosis analysis. Prognosis and prediction are more related to the analytical approach, whereas the outcomes listed are more clinical or public health relevant. However, some of these outcomes will need prognosis or prediction machine learning applications.

Q6: The sentence “This scoping review focuses on the applications of machine learning techniques in LMICs developed using LMICs data as prediction models and other algorithms work better, i.e. have better accuracy, in the population – data – for which they have been developed. Conversely, when these models are applied to other populations, settings or data, they need some modification (e.g., re-calibration).”, is not understandable.

A6: These lines have been modified to: “*This scoping review focuses on the applications of machine learning techniques in LMICs; these applications should have been developed using LMICs data, because prediction models and other algorithms work better, i.e. have better accuracy, in the populations – data – for which they were developed*”.

Q7: In eligibility criteria, authors don't mention the publication period of the studies screened. If there is no restriction on the date, a study should be published, this should be clearly reported. Authors don't also mention the dates that their search will take place.

A7: There will be no time restrictions, as it was reported in the ‘information sources’ section: “*These sources will be used without language or time/year restrictions*”. the date of the searches will be reported in the full scoping review as per the gold standard PRISMA-ScR reporting guidelines. This information is usually not included at the protocol stage.

Q8: Please revise what machine learning techniques are: Computer vision systems, computer-assisted image processing and natural language processing, are considered by the authors as machine learning techniques, but they actually are broader scientific fields.

A8: There are indeed broader scientific fields. Over the years these methods have evolved and became strong enough to be a scientific field themselves. However, they still use methods of machine learning as per our initial definition (see introduction, first paragraph): *to learn from data rather than from predefined instructions*.^{1,2}

Q9: The authors do not report the reasons behind the use of selected reporting tools, i.e., PRISMA-P, Levac, PRISMA-ScR. Proper justification is needed for the selection of those tools.

A9: PRISMA-P is a standard method/check list to report review protocols; similarly, PRSIMA-ScR is a well-known instrument to guide the reporting of scoping reviews. The methodology proposed by Levac is useful for scoping reviews that aim to generate a broad picture of the available evidence on a subject. These remarks have been included in the manuscript (‘Overview’ sub-heading).

Review #4

Q1: The definition of machine learning (ML) “analytical techniques through which computers learn directly from data, examples and experiences, rather than from a pre-programmed rule” is good. However distinguishing ML from more conventional statistical methods could be challenging and in

¹ Davies, S.C. “Annual Report of the Chief Medical Officer, 2018 Health 2040 – Better Health Within Reach”: Department of Health and Social Care (2018).

² Rebala G, Ravi A, Churiwala S. An Introduction to Machine Learning: Springer International Publishing; 2019.

some cases the definitions are rather unclear. For example regression analysis could be used as a machine learning approach but in some cases might be considered a pre-programmed rule.

A1: We appreciate the positive feedback and agree that in some cases the distinction may not be evident. However, from the context of the scientific paper, from the overall methodological approach, or from the aims, we will be able to reckon whether the study uses machine learning techniques versus a “simple” linear regression in the context of a standard statistical analysis. Moreover, we would reach out to authors for further information in case of uncertainty around the eligibility of a study. These arguments have been elaborated in the ‘Eligibility criteria’ section: *“Distinguishing between ML applications and more conventional statistical methods could be challenging because in some cases the definitions are unclear; for instance, regression analysis. Nonetheless, from the context of the scientific paper, from the aims or overall methodological approach, it is possible to reckon whether a study uses ML techniques versus more conventional statistical methods. If needed, we would reach out to authors for further information”*.

Q2: p6, l10 - Arguably classification and clustering are the same processes.

A2: Certainly. However, we could also argue that classification is based on supervised methods, whilst clustering may be based on unsupervised methods. For example, a classification model can learn from a dataset with a clear outcome variable; on the other hand, a clustering model can be developed to find groups (clusters) learning from the data without a pre-specific common pattern. In conclusion, there could be an overlap among different ML tasks or processes; this has been addressed in the ‘Introduction’ section (for further details please refer to the second comment by Reviewer #1).

Q3: Table 1. lists a number of terms, but is not comprehensive, authors should also consider regression analysis, gradient boosting, genetic algorithms, naïve Bayes, ridge regression, the list goes on...

A3: We could argue that the suggested new terms would be included in those listed in Table 1. For example, regression analysis will fall within the ‘predictive analytic’ term. Nonetheless, we do admit that the list is not comprehensive, and will acknowledge so in the limitations section of our final report.

Q4: Have the authors considered a specific review of clinical trials performed in LMICs?

A4: Thank you for your suggestion. At this point we aim to, first, describe the overall scenario of health and machine learning in low- and middle-income countries. Later, and based on the findings, we plan to conduct more focused systematic review addressing a specific question such as the effectiveness of machine learning in LMIC and including pertinent study designs.

Q5: “we will search Cochrane and Google Scholar (first 10 pages).” – why the first ten pages, this will surely not be a comprehensive approach.

A5: This is a common approach in scoping reviews.³ We will acknowledge this limitation in the final report.

³ Briscoe S. A review of the reporting of web searching to identify studies for Cochrane systematic reviews. Res Synth Methods. 2018;9(1):89–99. doi:10.1002/jrsm.1275

VERSION 2 – REVIEW

REVIEWER	George Gotsadze Curatio International Foundation
REVIEW RETURNED	12-Mar-2020

GENERAL COMMENTS	In my personal opinion, the authors have adequately addressed the issues raised by reviewers in the revised manuscript.
---

REVIEWER	Andreas Triantafyllidis Information Technologies Institute, Centre for Research and Technology - Hellas, Greece
REVIEW RETURNED	03-Apr-2020

GENERAL COMMENTS	All comments have been addressed
----------------------------------

REVIEWER	Michael Barnes Queen Mary University of London
REVIEW RETURNED	07-Apr-2020

GENERAL COMMENTS	This is an interesting scoping review proposal which is quite clearly described. However as it currently stands I do not think it will be a comprehensive review, the authors have attempted to address this and I think it will represent a useful study, although somewhat limited in scope.
--

VERSION 2 – AUTHOR RESPONSE

Reviewer #2

We appreciate you found our answers adequate.

Reviewer #3

We appreciate you found the revised version in better shape.

Review #4

We appreciate your constructive criticism and positive comments about our work.